# The molecular components of the anti-inflammatory cholinergic pathway are extrasplenic

Warda Merchant, Steven Wyler, Bandy Chen, Laurent Gautron *

Center for Hypothalamic Research and Department of Internal Medicine, UT Southwestern Medical Center, Harry Hines blvd, Dallas, Texas, Unites States of America

* Laurent.Gautron@UTSouthwestern.edu

## Abstract

The anti-inflammatory cholinergic pathway describes the interaction between cholinergic vagal nerves and splenic immune cells, yet the exact mechanisms underlying the anti-inflammatory cholinergic pathway remain disputed. Here, we mapped the expression of key molecular components of the anti-inflammatory cholinergic pathway in the adult mouse using RNAScope in situ hybridization (ISH) and quantitative PCR (qPCR). In C57BL/6J wild-type male mice, we observed the expression of choline acetyltransferase (*Chat*) and alpha 7 nicotinic acetylcholine receptor (*Chrna7*) in various autonomic neurons throughout the body, but not in the spleen, even after bacterial lipopolysaccharide (LPS) treatment. In contrast, the beta-2 adrenergic receptor (*Adrb2*), another autonomic receptor with well-documented anti-inflammatory actions, was highly expressed in the spleen, with a significant decrease following LPS administration. Interestingly, *Adrb2* was also expressed at lower levels in the spleen of a newly generated global knockout mouse for *Chrna7*. Lastly, we did not observe YFP-positive cells or axons in the spleen of the ChAT-Cre-ChR2-YFP mouse. Based on our findings, we propose a new model of the cholinergic anti-inflammatory pathway that highlights the roles of extrasplenic cholinergic signaling.

## Introduction

The electrical stimulation of the vagus nerve (VNS) inhibits the release of pro-inflammatory cytokines in laboratory rodents [1–8]. To explain these anti-inflammatory effects, it has been proposed that electrically stimulated vagal motor neurons trigger nerves reflexes which, subsequently, induce the acute release of acetylcholine (ACh) within the spleen [6,9–13]. The release of ACh in the latter structures has been attributed to a subtype of choline acetyltransferase (*Chat*)-expressing immune cells sensitive to changes in autonomic outflow [6,14–16]. Ultimately, binding of immune-derived ACh to the alpha 7 nicotinic acetylcholine receptor (α7nAChR)

**Data availability statement:** All relevant data are within the manuscript.

**Funding:** LG: NIH P01DK119130 (CNS mechanisms linking exercise training with energy balance and metabolism, Core C). LG: TRC4 #066 (Physiological and molecular determinants of shock-induced torpor in the mouse).

**Competing interests:** The authors have declared that no competing interests exist.

expressed by immune cells dampens the release of proinflammatory cytokines [6,10–13,17–19]. In agreement with the above scenario, the α7nAChR knockout mouse model resists the anti-inflammatory actions of VNS [19]. In addition, quantitative PCR (qPCR) and RT-PCR studies have reported *Chrna7* (transcript for α7nAChR) expression in lymphoid tissues and isolated immune cells in rodents and humans [10,19–27]. Immune cells bearing α7nAChR have also been detected using techniques such as GFP reporter mice, flow cytometry antibodies, and alpha-bungarotoxin binding [10,28,29,19]. Lastly, the administration of α7nAChR agonists inhibits the release and expression of pro-inflammatory cytokines both in vivo and in vitro [30–39]. Hence, the vast majority of the review articles on the anti-inflammatory actions of VNS, often referred to as the anti-inflammatory cholinergic pathway, describe α7nAChR-expressing splenocytes as an obligatory link between the vagus nerve and immunity [40–47].

On the other hand, α7nAChR agonists do not always relieve inflammation in laboratory rodents depending on the examined inflammatory paradigm [48–50]. Of note, one clinical trial did not confirm the anti-inflammatory effects of a α7nAChR agonist in human subjects acutely treated with bacterial lipopolysaccharides [51]. Moreover, the exact mechanisms underlying the anti-inflammatory cholinergic pathway remain disputed. For instance, no anatomical and functional connections between vagal motor neurons and the spleen exist in the rat [52]. Others noted that electrically stimulated vagal motor neurons can inhibit inflammation in the α7nAChR knockout model [53,54]. Adding to the latter concerns, the anti-inflammatory effects of commonly used α7nAChR agonists are largely intact in primary macrophages from α7nAChR knockout mice [26], in the presence of α7nAChR antagonists [55], and in splenectomized animals [56]. Moreover, the administration of α7nAChR agonists exerts beneficial effects in α7nAChR knockout mice suffering from sepsis [57]. It is therefore likely that α7nAChR agonists exert α7nAChRs-independent anti-inflammatory effects.

More importantly, whether immune cells express α7nAChR remains debated. While prior studies reported α7nAChR immunoreactivity in immune and splenic cells [20,29,58–61], the detection of α7nAChR using commercial antibodies is deemed unreliable by many experts [62–66]. Whereas numerous reports of *Chrna7* mRNA expression in immune cells [10,19–26], other studies found very low levels of *Chrna7* mRNA in unstimulated human immune cells and cell lines [22,67,68]. One recent study failed to detect *Chrna7* in the mouse spleen using RNA sequencing [69]. By in situ hybridization (ISH), another study similarly failed to detect *Chrna7* in the developing mouse spleen [70]. Given the major discrepancies listed above, this study aimed at reassessing the *in vivo* distribution of the two main molecular components of the anti-inflammatory cholinergic pathway: *Chat* and *Chrna7*. In parallel, we also assessed the expression of beta-2 adrenergic receptor (*Adrb2*), another autonomic receptor with well-documented anti-inflammatory effects [11,50,71–74].

## Methods

Experiments listed below were approved by the University of Texas Southwestern Institutional Animal Care and Use Committee under protocols 2016–101605 and 2017–101994. Mice were given an overdose of Ketamine/Xylazine (500/50 mg/kg,

i.p.) before being transcardiacally perfused. All the mice were bred and studied at the Animal Resource Center at The University of Texas Southwestern Medical Center. Mice were grouped-housed in ventilated cages with enrichment (nestlet and igloo) in a light (12 h on/12 h off; lights on at 6 am)- and temperature-controlled environment (21.5°C–22.5°C). Water and standard chow (Harlan Teklad TD.2016 Global) were provided without restrictions. The number of mice used for experiments is indicated below. Anesthesia and/or analgesia were not provided since all our procedures were terminal. Mice treated with LPS were always sacrificed within a few hours to alleviate suffering.

### Commercial mice

Male mice on a C57Bl/6J background were ordered from the Jackson Laboratory (stock #000664) and animals were used when they reached approximately 6–8 weeks of age. One cohort was used for RNAscope studies (n = 4 saline; n = 4 LPS). Another cohort of mice treated with either saline (n = 5) or LPS (n = 5) was used for quantitative real-time PCR (qPCR) as described below. An additional cohort of mice was treated with a high dose of LPS (n = 5) for a qPCR control experiment. In addition, we generated male Chat-Cre-Chr2-YFP mice by crossing ChAT-Cre mice from the Jackson Laboratory (strain #:031661) with Ai32 (strain #:024109). One cohort was used for RNAscope studies (n = 4 saline; n = 4 LPS).

### Generation of Chrna7 knockout mice

To generate a conditional, Chrna7 loxP- flanked allele (Chrna7 flox), we used CRISPR-Cas9-mediated homologous recombination to insert two LoxP sites flanking exon 1 and 2 of the Chrna7 gene encoding the transcriptional and translational start site as well as the first 65 amino acids of nAChRα7. Guides were designed using IDT's design algorithm as follows: 5' Guide ribonucleotides sequence: 5'- AGGACCCACGGUCAAUUUUCGUUUUAGAGCUAUGCU −3' 3' ribonucleotide sequence: 5'- UUAGAAAACAAUCGUCCACGGUUUUAGAGCUAUGCU-3'. Two IDT Ultramers™ containing 80 bp homology arms flanking the loxP sequence were used as the HDR templates. Guides, trcRNA, HDR templates and Cas9 protein (all from Integrated DNA Technologies, Coralville, IA) were administered through a pronuclear injection in fertilized C57Bl/6NCrl (Charles River Laboratories, Wilmington MA) zygotes by the UT Southwestern Transgenic Technology Center. Founders were screened by PCR and sanger sequencing. Germline transmission of the floxed allele was verified by crossing Founders to C57Bl/6NCrl mice. A floxed mouse was crossed with a germline, CMV-Cre mouse (Schwenk et al 1995 PMID: 8559668) to generate knockout mice. This mouse was subsequently crossed to a C57Bl/6NCrl to "breed-out" the CMV-Cre allele. Mice were genotyped using the following primers: 5'-CACTGCATGTGATCCTGAAGAG-3', 5'-AAGTGAAAAGCAAGGCTGGAG-3', and 5'-AGGGAAGCCAAACCTCAAGATG-3' with a wildtype band of 123 bp, a floxed band of 163 bp, and a knockout band of 430 bp. Wild-type (n = 6), heterozygous (n = 8), and homozygous (n = 8) were used for the purpose of qPCR assays.

### LPS treatment

LPS (Sigma L2880 Escherichia coli 055:B5) was prepared in sterile pyrogen-free 0.9% saline (Hospira, IL, USA). A single dose of LPS at 1 mg/kg of body weight (i.p.) was administered during the light phase and animals were sacrificed 2 hrs post-injection. The dose and time of sacrifice were previously considered suitable for studying the vagal anti-inflammatory pathway in mice [75]. Notably, a separate cohort was treated with a high dose of LPS (15 mg/kg, i.p.) for a qPCR control experiment.

### QPCR

Analysis of gene expression by quantitative real time PCR (qPCR). Total mRNA was isolated from brainstem, duodenum, and spleen using RNA PureZol (Bio-Rad, Inc). RNA concentration was assessed by UV spectroscopy using the Beer-Lambert law at 260 nm wavelength. Complementary DNA from 1 µg of input RNA was generated with the

High-Capacity cDNA Reverse Transcription Kits (Life Technologies). cDNA was diluted in ultra-pure distilled water, DNase-free RNase free (Invitrogen, 10977015). mRNA transcript levels were measured in duplicate using TaqMan Universal PCR Master Mix (Applied Biosystems, 4304437), the ABI PRISM 7900HT Sequence Detection System (Applied Biosystems, 4317596) and the CFX maestro software. Prevalidated Taqman assays were performed using primers of Mm01317884_m1 (Chrna7), Mm01221876_m1 (Chat), and Mm02524224_s1 (Adrb2). Relative gene expression was calculated by the 2-ΔΔCT method with 18s as a control gene.

## RNAscope studies

The protocol used below follows recommendations from our prior publication [76]. Mice were given an overdose of Ketamine/Xylazine (500/50 mg/kg, i.p.) before being transcardiacally perfused with 1x phosphate buffer saline (PBS) and 10% formalin (Sigma). Samples of interests were collected, kept in formalin at 4°C for 24 hours, and switched to 30% sucrose for another 24 hours. Samples were next positioned inside a drop of OCT medium (Sakura), frozen on dry ice, and stored at −80°C. Samples were cut with a cryostat into sections of 14 µm thickness and collected on SuperFrost glass slides. Pretreatment and ISH following the recommendations from the manufacturer (Advanced Cell Diagnostics, USA). Briefly, slides were rinsed, baked, post-fixed, and dehydrated in series of ethanol. Subsequently, slides were treated with $H_2O_2$, target retrieval, and protease plus as recommended by the manufacturer. Different combinations of probes for *Adrb2* (#449771), *Chat* (#408731-C2), *Phox2b* (#407861-C2), and *Chrna7* (#465161) were applied for 2 hours at 40°C (HybEZ oven). Slides were next incubated with amplification reagents contained in the multiplex kit (cat# 323100) and appropriate Opal dyes 520, 570, 690 (1/1,500; Akoya Biosciences). Slides were rinsed, exposed to DAPI, before applying EcoMount (BioCare Medical, USA) and coverslip. Other spleen slides were incubated with probes for *Adrb2* (#449771), *Chat* (#408731), or *Chrna7* (#465161) and processed with a FastRed chromogenic assay (ACD #323910) following the manufacturer's instructions. Hematoxylin counterstained was applied to the latter slides before applying mounting medium and coverslip.

## GFP immunostaining

A standard procedure was used to label YFP after RNAscope staining. Briefly, tissues already treated with RNAscope reagents were incubated with an antiserum against GFP (Aves Lab., cat. no. GFP-1020). This is a chicken polyclonal antiserum raised against GFP emulsified in Freund's adjuvant. We are confident in the specificity of this antiserum for several reasons: a) according to the manufacturer, this antibody detected a single 28 kDa band using western blot analysis; b) this antibody was previously shown to produce no staining in the tissues of wildtype mice [77]; c) The staining obtained in this study was highly consistent with the anticipated distribution of GFP in our samples. Following incubation with the primary antibody, slides were incubated with a biotinylated anti-chicken secondary (Jackson ImmunoResearch, 1:1,000 dilution) and streptavidin AlexaFluor488 (Invitrogen, 1:1,000 dilution). Slides were rinsed, exposed to DAPI, before applying EcoMount and a coverslip.

## Microscopic and data analysis

Digital images of FastRed-stained spleens were captured with an epifluorescence microscope (Leica DM6 B microscope with Leica DFC 9000 GT digital microscopy camera) attached to the LAS X software. Digital fluorescent images used for plates were acquired by confocal microscopy using the Zeiss (LSM880 Airyscan) under standardized acquisition settings. A typical image was generated with stacks of optical sections. Image J Fiji was used to merge our final RGB images at 300 dpi. For nodose ganglion sections, we manually estimated the proportion of Chrna7-expressing profiles expressing Phox2b mRNA. The number of positive Chrna7 dots per profile was also manually counted on digital images using the multi-points command. Adobe Photoshop 2021 was used to generate final plates and make annotations. We slightly adjusted the contrast and brightness of images uniformly. All graphs were generated, and statistical analyses were

performed using GraphPad Prism 10 with data presented as mean values±S.E.M. The number of mice per group and the statistical results are provided in the Fig legends and/ or result section.

## Statistical reporting

Following a between subjects experimental design, animals were randomly attributed to treatment groups (saline or LPS). The experimenter performing qPCR was not aware of experimental groups. Once experimental data were collected, however, an unblinded experimenter performed the statistical tests. Raw data were not transformed, and no outliers were removed. Data in Fig 1 were analyzed by unpaired t-test (normal distribution) to compare saline vs LPS groups. Calculation was two-tailed. Data in Fig 5 with more than two groups were analyzed by ordinary one-way ANOVA (not matching or pairing with normal distribution). Multiple comparison was accounted for with the Dunnett test (recommended by GraphPad). Based on conventional practices and literature in qPCR studies, the threshold for statistical significance is typically set at a p-value of less than 0.05 (p<0.05).

## Results

### QPCR analysis of the main molecular components of the anti-inflammatory pathway in wild-type mice

We examined by qPCR the expression for *Adrb2*, *Chat*, and *Chrna7*, three molecules known to be involved in the regulation of immunity by the autonomic nervous system. We focused on the brainstem compared to structures linked to the autonomic control of immunity, with a special emphasis on the gastrointestinal tract (GI) tract and spleen, in animals treated with either saline or LPS (Fig 1). *Adrb2* expression was moderate in the brainstem and duodenum, and high in the spleen (Fig 1A). Interestingly, LPS treatment significantly reduced *Adrb2* in the brainstem (unpaired t test; p=0.0276, t=2.687, df=8) and spleen (p=0.0250; t=2.753, df=8), but not in the duodenum (p=0.6640, t=0.4509, df=8) (Fig 1A). *Chrna7* expression was expressed at moderate levels in the brainstem and not regulated by LPS (unpaired t test; p=0.9452, t=0.07095, df=8) (Fig 1B). In the duodenum, *Chrna7* expression was low and not regulated by LPS (p=0.3811, t=0.9269, df=8). In the spleen, *Chrna7* expression was very low, close to its limits of detection (Ct

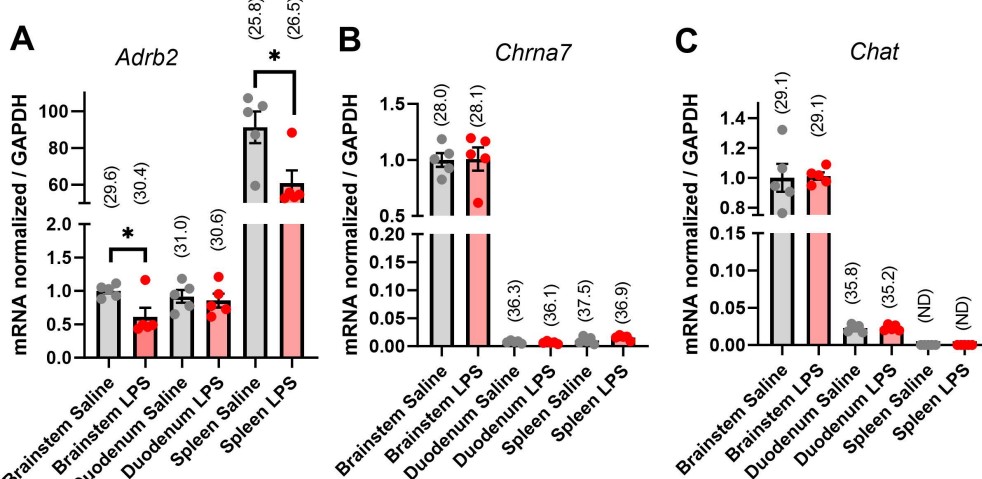

**Fig 1. QPCR analysis of *Adrb2*, *Chrna7*, and *Chat* expression in wild-type mice.** Samples were obtained from mice treated with saline (grey dots) or LPS (red dots; n=5 per group). LPS was administered at 1 mg/kg (ip) and mice sacrifice 2 hours post-injection. Gene expression was normalized against GADPH. Data are expressed as mean fold changes values±S.E.M. Average Cq values are indicated above each bar graph. **(A)** *Adrb2* expression in the brainstem, duodenum, and spleen of saline- and LPS-treated mice. *, p=0.0276 (unpaired t test). **(B)** *Chrna7* expression in the brainstem, duodenum, and spleen of saline- and LPS-treated mice. **(C)** *Chat* expression in the brainstem, duodenum, and spleen of saline- and LPS-treated mice.

approaching 37), and not regulated by LPS (p = 0.1454, t = 1.613, df = 8) (Fig 1B). *Chat* expression was expressed at moderate and low levels in the brainstem and duodenum, respectively (Fig 1C). LPS did not regulate *Chat* in the brainstem (unpaired t test; p = 0.9040, t = 0.1245, df = 8) and duodenum (p = 0.8479, t = 0.1981, df = 8). *Chat* remained completely undetectable in the spleen. We wondered if a very high dose of LPS could influence *Chat* expression. To address this point, we conducted additional qPCR experiments using a high LPS dose (15 mg/kg, i.p.; S1 Fig). Notably, *Chat* remained undetectable regardless of the dosage.

## Expression of Chrna7 is restricted to neurons

To validate our qPCR data and clarify α7nAChR target cells, we used fluorescent RNAScope assays to survey the distribution of *Chrna7* mRNA in the brainstem, gut and spleen (Fig 2). Abundant *Chrna7* signals (red dots) accumulated in neurons throughout the parasympathetic nervous system including vagal motor neurons (Fig 2A), nodose-jugular ganglion neurons (Fig 2B, 2C), and the enteric nervous system (Fig 2D, 2E). In contrast, *Chrna7* signals were virtually undetectable in the GI mucosa (Fig 2D) and Peyer's patches (Fig 2F). Within the sympathetic nervous system, many neurons belonging to the dorsal root ganglion (Fig 2G) and celiac ganglion (Fig 2H) also expressed *Chrna7* at high levels. The mouse spleen was devoid of *Chrna7* signals (Fig 2I).

Multiplex fluorescence RNAScope further revealed that *Chrna7* was expressed by at least 50% of Phox2b-positive vagal afferent neurons (Fig 3A–3D). The administration of LPS did not alter the overall distribution and percentage of *Chrna7*-expressing neurons (Fig 3E). To ascertain the absence of *Chrna7* and *Chat* in the spleen, we also performed chromogenic RNAscope assays, a method suited to detecting lowly expressed transcripts and avoid issues of endogenous fluorescent background. As anticipated, *Adrb2* signals (red dots) were observed across the entire mouse spleen in both the white and red pulps (Fig 4A, 4B). By contrast, we could never detect any signals for either *Chrna7* or *Chat* in the spleen (Fig 4C–4F).

## Transcriptional adaptations in the spleen of α7nAChR knockout mice

The expressions of *Chrna7*, *Chat* and *Adrb2* were further assessed in a newly generated knockout mouse (Fig 5A). Expression for *Chrna7* was significantly reduced in the duodenum of mice lacking one copy of *Chrna7* and undetectable in the knockout mouse (One-way ANOVA, F = 20.63, P < 0.0001; followed by a Dunnett post-hoc comparison with **** for p < 0.0001) (Fig 5B). In the spleen of wild-type and heterozygous animals, *Chrna7* expression was detected in several samples, albeit at very low levels (One-way ANOVA, F = 2.208, P = 0.1360; ns, not significant). (Fig 5C). Since *Chrna7* expression was not significantly different between genotypes, we concluded that low levels of *Chrna7* in the whole spleen are attributable to a very small population of *Chrna7*-expressing cells that are undetectable by histology.

Interestingly, *Chat* was significantly reduced in the duodenum of mice lacking one or two copies of *Chrna7* (One-way ANOVA, F = 16.02, P < 0.0001; followed by a Dunnett post-hoc comparison with *** for p < 0.001) (Fig 5D), suggesting adaption of the autonomic nervous system in response to α7nAChR deficiency. In agreement with our prior findings, *Chat* transcripts couldn't be detected in the spleen (Fig 5E). Lastly, we report an unexpected reduction of *Adrb2* expression in both the duodenum (One-way ANOVA, F = 8.924, P = 0.0019; followed by a Dunnett post-hoc comparison with ** for p < 0.01) and spleen (One-way ANOVA, F = 7.995, P = 0.0033; followed by a Dunnett post-hoc comparison with ** for p < 0.01) of mice lacking *Chrna7* (Fig 5F, 5G).

## The spleen is not a site of cholinergic signaling

To clarify the relationship between cholinergic cells and *Chrna7*-expressing neurons, multiplex RNAscope for *Chrna7* was repeated in a reporter mouse model with permanently labeled cholinergic cells (ChAT-Cre-ChR2-YFP). In these mice, YFP was seen in vagal motor neurons and their axons traveling in the vagus nerve (Fig 6A, 6B), as well as in enteric neurons and their axons throughout the mucosa (Fig 6C, 6D). Within the mucosa, we observed round shape cells resembling

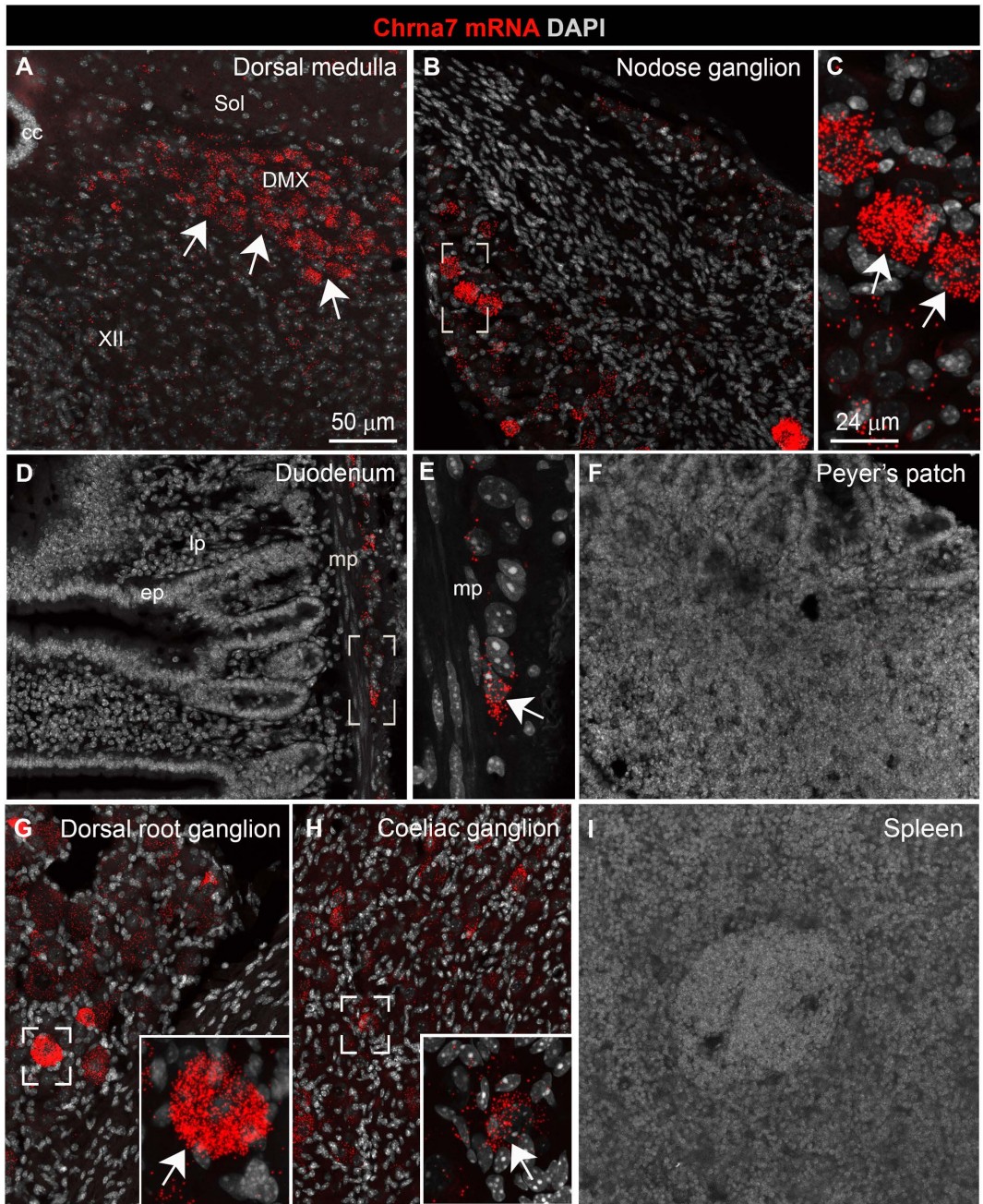

**Fig 2. Anatomical distribution of *Chrna7* in wild-type mice.** Hybridization signals for *Chrna7* (red dots) were found in neurons throughout the para-sympathetic nervous system including the dorsal nucleus of the vagus **(A)**, the nodose ganglion **(B, C)**, and the enteric nervous system **(D, E)**. White arrows indicate representative neuronal profiles positive for *Chrna7*. Note that Chrna7 expression was not seen in non-neuronal areas such as the intestinal mucosa **(D)** or the Peyer's patch **(F)**. Hybridization signals for *Chrna7* were also abundant in neurons belonging to the dorsal root ganglion **(G)** and celiac ganglion **(H)**, but not the spleen **(I)**. white arrows indicate representative *Chrna7*-expressing neurons. Tissues were counterstained with DAPI (grey) and all images were acquired using confocal microscopy. Abbreviations: cc, central canal; DMX, dorsal nucleus of the vagus; ep, epithelium; lp, lamina propria; mp, myenteric plexus; Sol, nucleus of the solitary tract; XII, hypoglossal nucleus. Scale bar in A applies to B, D, F, G, **I.** Scale bar in C applies to E and the insets in G and **H.**

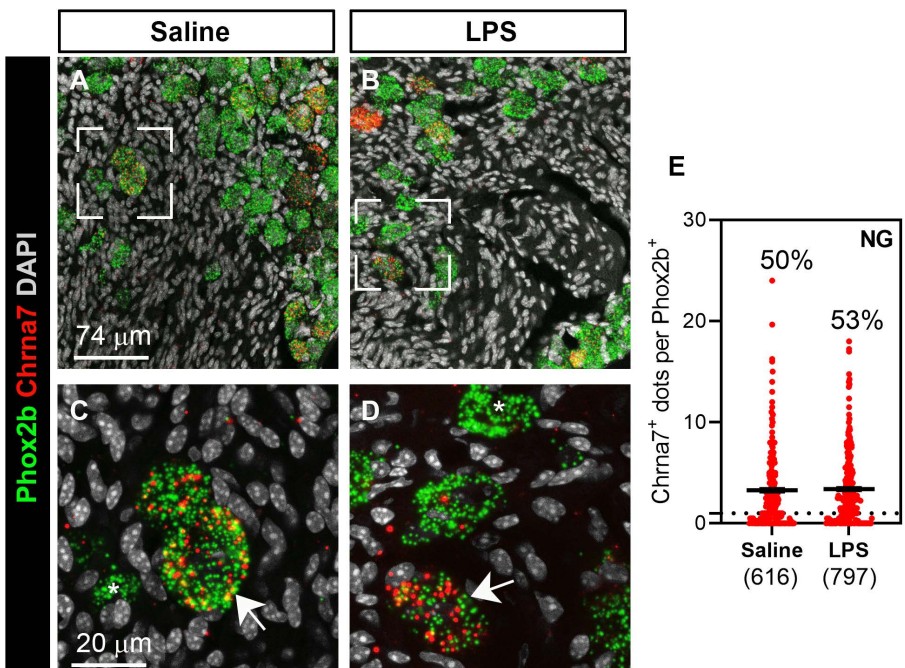

**Fig 3. *Chrna7* expression is unchanged in the nodose ganglion of LPS-treated mice.** Multiplex RNAScope for *Phox2b* (green dots) and *Chrna7* (red dots) in the nodose ganglion of saline- (**A, C**) and LPS-treated (**B, D**) wild-type mice (n = 4 mice/group). White arrows indicate representative neurons co-expressing both transcripts. White asterisks indicate neurons solely positive for *Phox2b*. A visual inspection of the ganglia did not reveal obvious differences in the intensity of signals or frequency of *Chrna7*-positive neurons. Tissues were counterstained with DAPI (grey) and all images were acquired using confocal microscopy. **(E)** Graphs showing the *Chrna7* signal strengths after saline or LPS treatments. The number of *Chrna7*-positive dots per identified *Phox2b*-positive profile was counted. Each dot represents 1 identified profile and the total number of counted *Phox2b* profiles is listed below each group. The black horizontal lines represent the mean numbers of dots per profile ± SEM. The percentage of *Chrna7*-positive Phox2b cell is indicated above each plot. Abbreviations: NG, nodose ganglion.

immune cells in the GI mucosa (Fig 6D). However, these cells were only occasionally observed and not consistently in every animal. A similar observation was made in the Peyer's patch (Fig 6E). As expected, YFP also labeled cholinergic terminals in postganglionic sympathetic ganglia such as the celiac ganglion (Fig 6F). Overall, YFP staining was distributed a pattern good agreement with the known distribution of cholinergic cells throughout the body [16]. However, neither YFP-stained cells nor axons were observed in the mouse spleen (Fig 6G). Observations were repeated in a total of 4 saline- and 4 LPS-treated mice. However, the above distribution pattern was strictly identical in both treatment groups.

We next verify whether YFP cells were genuinely cholinergic, the GI tract and spleen of YFP mice were processed for multiplex RNAscope for *Chat* and *Chrna7* mRNAs. In the GI tract, most YFP-positive enteric neurons expressed *Chat* signals (Fig 7A–7D). A large subset, but not all of them, also expressed *Chrna7* signals. The proportions of *Chat*- and *Chrna7*-expressing YFP enteric neurons were directly comparable in saline- and LPS-treated mice (Fig 7D). Once again, in the spleen of the same animals, YFP was not detected and signals for *Chat* and *Chrna7* were completely absent (Fig 7E–7G).

## Discussion

Many previous studies have reported detecting α7nAChR expression in immune cells using mRNA-based techniques, binding assays, and histology (see Introduction). However, the specificity of the latter data was not always convincingly established. *In vitro*, immunocompetent cells including, among other examples, cultured macrophages, microglia, and

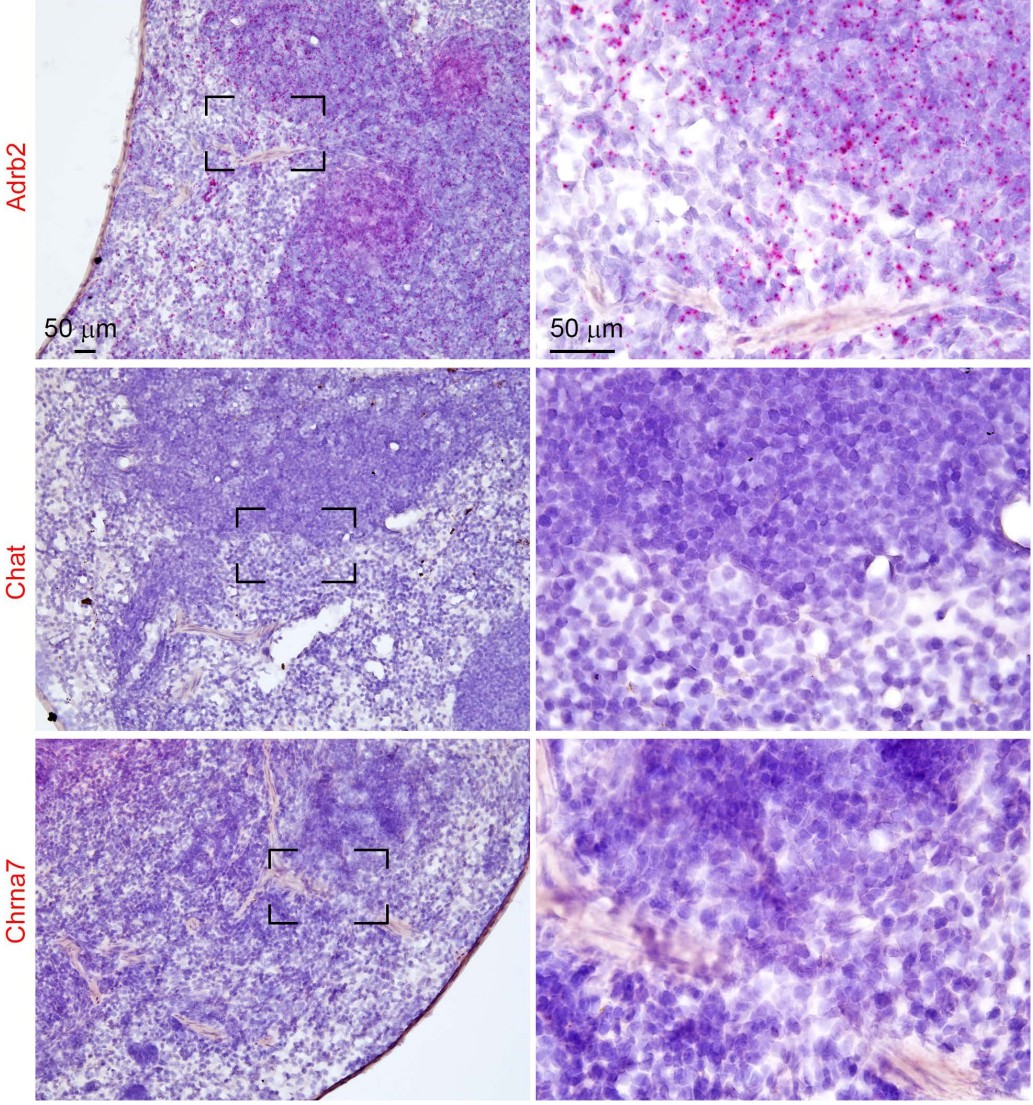

**Fig 4. Chromogenic RNAScope assay for *Adrb2*, *Chat* and *Chrna7* in the mouse spleen.** Chromogenic RNAScope (FastRed dots) was performed on the spleen of wild-type mice. Tissues was counterstained with hematoxylin. **(A)** *Adrb2* positive signals were seen across most of the spleen in both the white and red pulps. **(B, C)** In agreement with our fluorescent assays, the spleen was virtually devoid of any signals for *Chat* and *Chrna7*.

isolated B-cells have been shown to express very low levels of *Chrna7* transcripts using qPCR [26,78,79]. *In vivo*, several studies failed to detect *Chrna7* in lymphoid tissues and immune cells in mice and humans [69,70]. Our present data combined with that of others show that *Chrna7* is abundantly expressed by autonomic and enteric neurons [66,80–82], but only at very low levels by splenic cells. Although low levels of *Chrna7* transcripts were detected by qPCR, RNAscope consistently failed to detect *Chrna7* in the spleen. Combined, our results indicate that there are very few *Chrna7*-expressing cells in the mouse spleen. Furthermore, LPS administration did not modify the distribution pattern and expression levels of *Chrna7*, at least at the dose and time point used in this study. Likewise, publicly available transcriptomics data show enrichment of β-2 adrenergic receptors, but not *Chrna7* in the mouse, human, and non-human primate spleen as well as

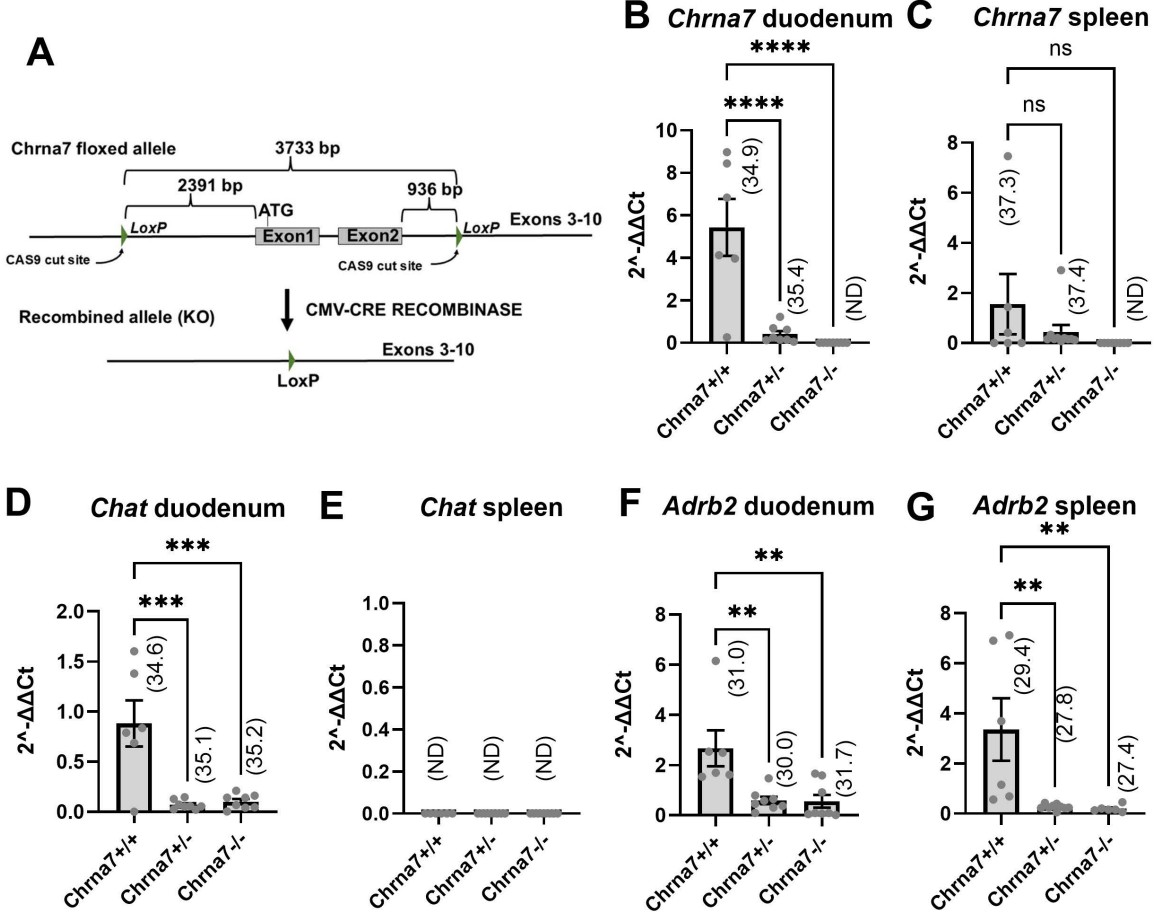

**Fig 5. Validation of a new Chrna7 knockout mouse. (A)** Schematic representation of the transgenes used to generate Chrna7 knockout mice. **(B)** QPCR for *Chrna7* expression in the duodenum from wildtype vs Chrna7+/– and Chrna7–/– mice (N = 6 – 9 mice per genotype). **** for p < 0.0001 (Dunnett post-hoc comparison). **(C)** QPCR for *Chrna7* expression in the spleen from wildtype vs Chrna7+/– and Chrna7–/– mice (ns, not significant). **(D)** QPCR for *Chat* expression in the duodenum from wildtype vs Chrna7+/– and Chrna7–/– mice. *** for p < 0.001 (Dunnett post-hoc comparison with). **(E)** QPCR for *Chat* expression in the spleen from wildtype vs Chrna7+/– and Chrna7–/– mice. Gene expression was undetectable (ND) in all samples. **(F)** QPCR for *Adrb2* expression in the duodenum from wildtype vs Chrna7+/– and Chrna7–/– mice. ** for p < 0.01 (Dunnett post-hoc comparison). **(G)** QPCR for *Adrb2* expression in the spleen from wildtype vs Chrna7+/– and Chrna7–/– mice. ** for p < 0.01 (Dunnett post-hoc comparison). Average Ct values for each group is listed above bar graphs.

varied immune cell types [83–85]. A publicly available transcriptional survey of blood-derived human immune cell types shows the absence of *Chrna7* and other cholinergic markers, even in activated cells [86]. These data from independent databases are recapitulated in Fig 8 and are in good agreement with each other's and our own findings. Herein, definitive evidence of *Chrna7* expression by immune cells is still lacking.

Prior studies have indicated that *Chat*-expressing lymphocytes modify immunity (15, 92, 93). However, whether immune cells are a significant source of ACh remains an ongoing debate [87]. By histochemical techniques, cells positive for cholinergic markers can be detected in the spleen of mice and humans even though at low density [88–90]. We and others have previously reported the presence of a few putative choline acetyltransferase (Chat)-positive cells in the mouse spleen using fluorescent reporter mice [6,14,16,23,91,92]. However, the visualization of ChAT-Cre or -GFP cells doesn't preclude that ACh is being produced by these cells. Moreover, ChAT stains were not validated using knockout model, leaving uncertainty as to its specificity. ACh release is stimulated within the rodent spleen after nerve stimulation [6,14,88].

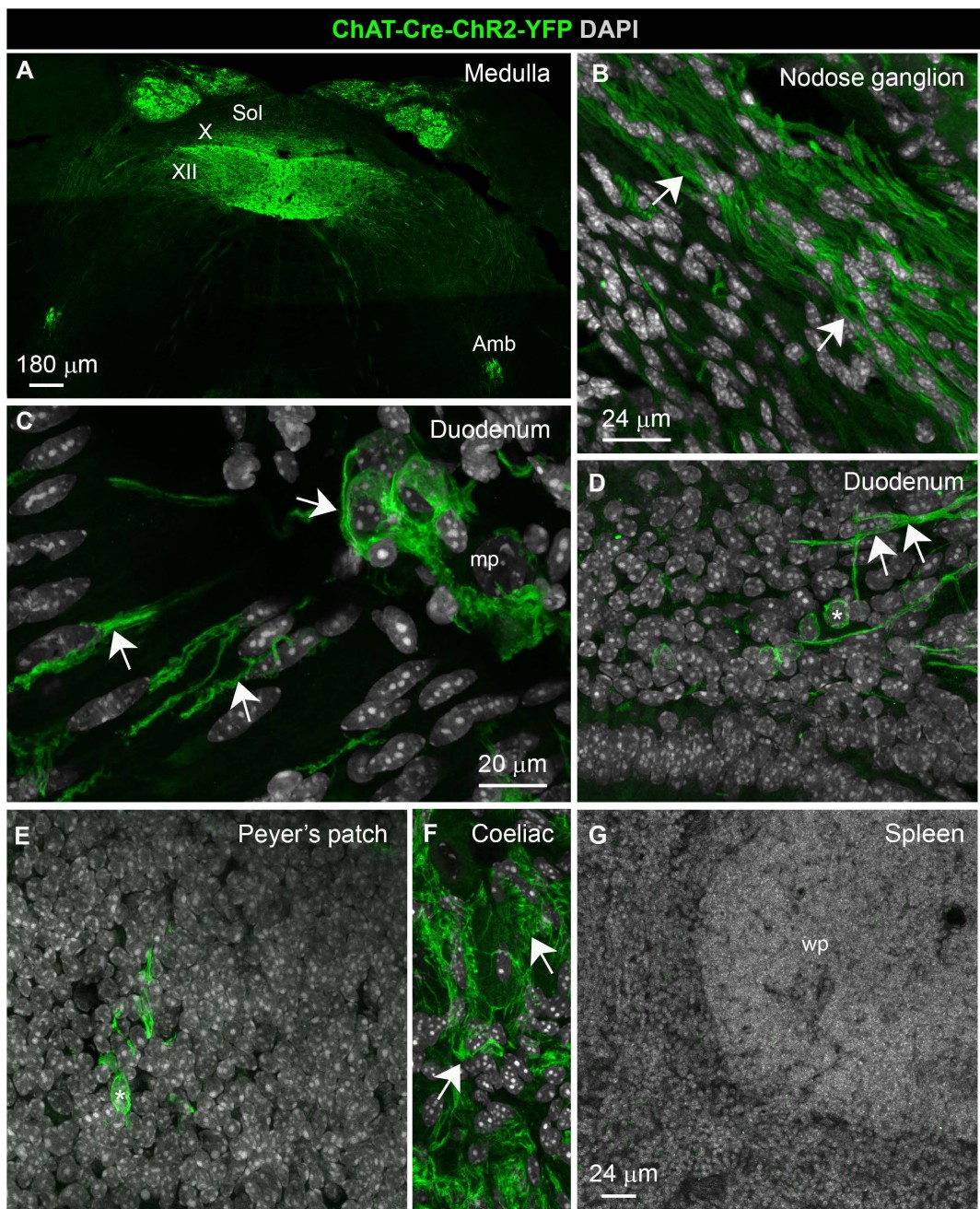

**Fig 6. Distribution of YFP in the Chat-Cre-ChR2-YFP mouse.** Confocal microcopy of YFP-stained tissues revealing the distribution of putative cholinergic structures in the ChAT-Cre-ChR2-YFP mouse. **(A)** In the brainstem, both YFP-positive somas and axons were observed in cholinergic nuclei containing vagal motor neurons. **(B)** Within the nodose ganglion, only YFP-positive axons were seen, likely originating from vagal motor neurons. **(C)** The GI tract contained abundant YFP immunoreactivity in enteric neurons, both submucosal and myenteric. The axons of enteric neurons traveled towards the mucosa. **(D)** Of note, the mucosa also contained round-shape YFP positive cells resembling immune cells (white asterisk), but only occasionally and not consistently across animals. **(E)** Sparse YFP-positive cells resembling immune cells were also occasionally detected in the Peyer's patches. **(F)** A rich network of YFP-positive axons was observed in postganglionic sympathetic ganglia, such as the celiac ganglion, but no cell bodies. **(G)** The spleen was devoid of YFP-positive axons and cells. Upon visual inspection, the above distribution pattern was strictly identical in saline- and LPS-treated mice. Abbreviations: mp, myenteric plexus; sol, solitary tract nucleus; X, vagus nerve; XII, hypoglossal nerve; Amb, nucleus ambiguous.

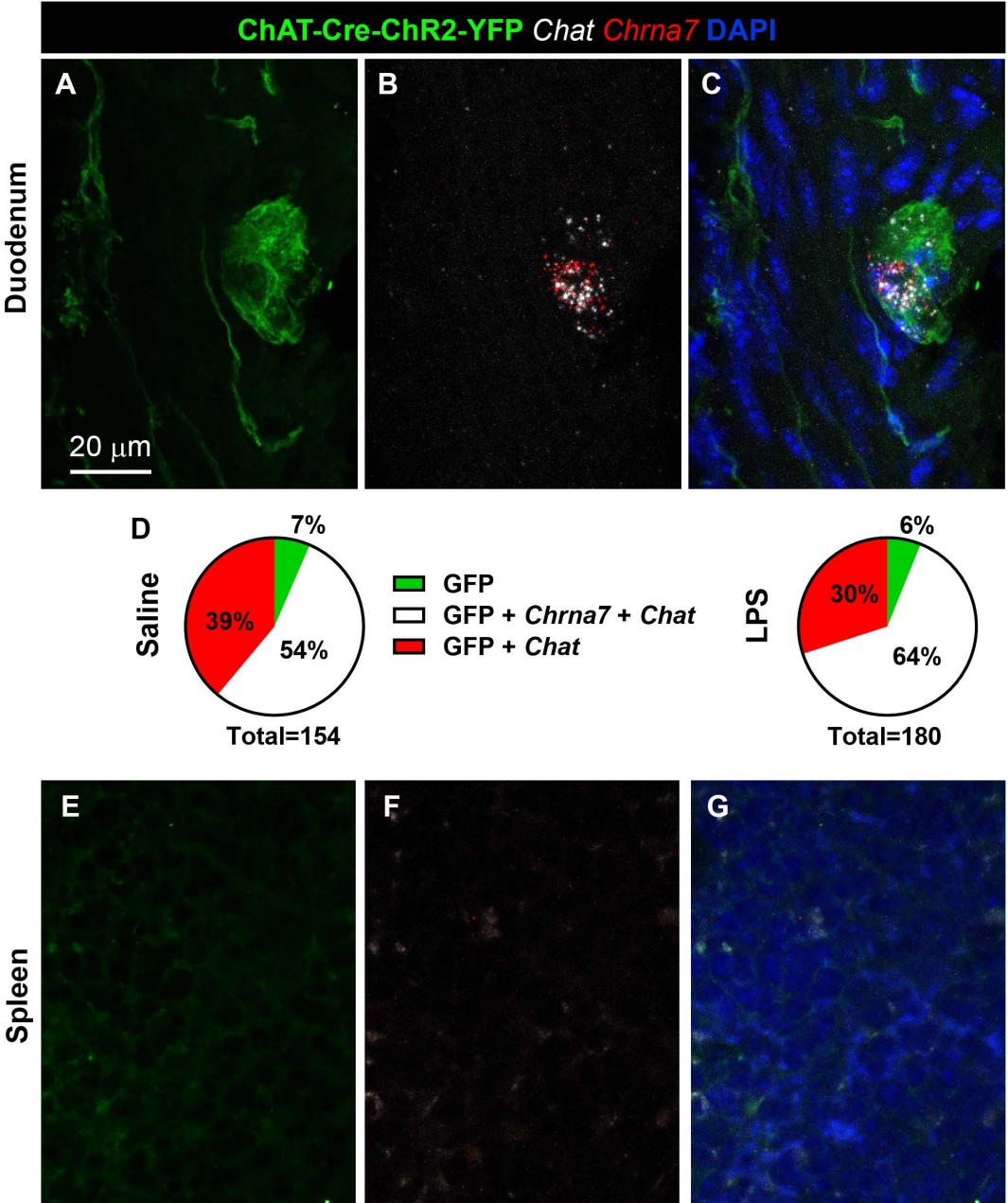

**Fig 7. Cholinergic signaling in the gut and spleen of ChAT-Cre-ChR2-YFP mice. (A-C)** Confocal microscopic analysis of RNAscope for *Chat* (white dots) and *Chrna7* (red dots) expression in the duodenum of ChAT-Cre-ChR2-YFP mice, with GFP-positive enteric neurons (green staining) identified using a GFP antibody. YFP-positive enteric neurons commonly co-expressed both transcripts. Tissue was counterstained with DAPI (blue). **(D)** Pie charts illustrating the percentage of GFP-positive cells in the duodenum that express *Chat* and/or *Chrna7*. **(E-G)** In the spleen of the same mice, neither YFP expression nor the transcripts for *Chat* and *Chrna7* were detected. Scale bar in **(A)** applies to all images.

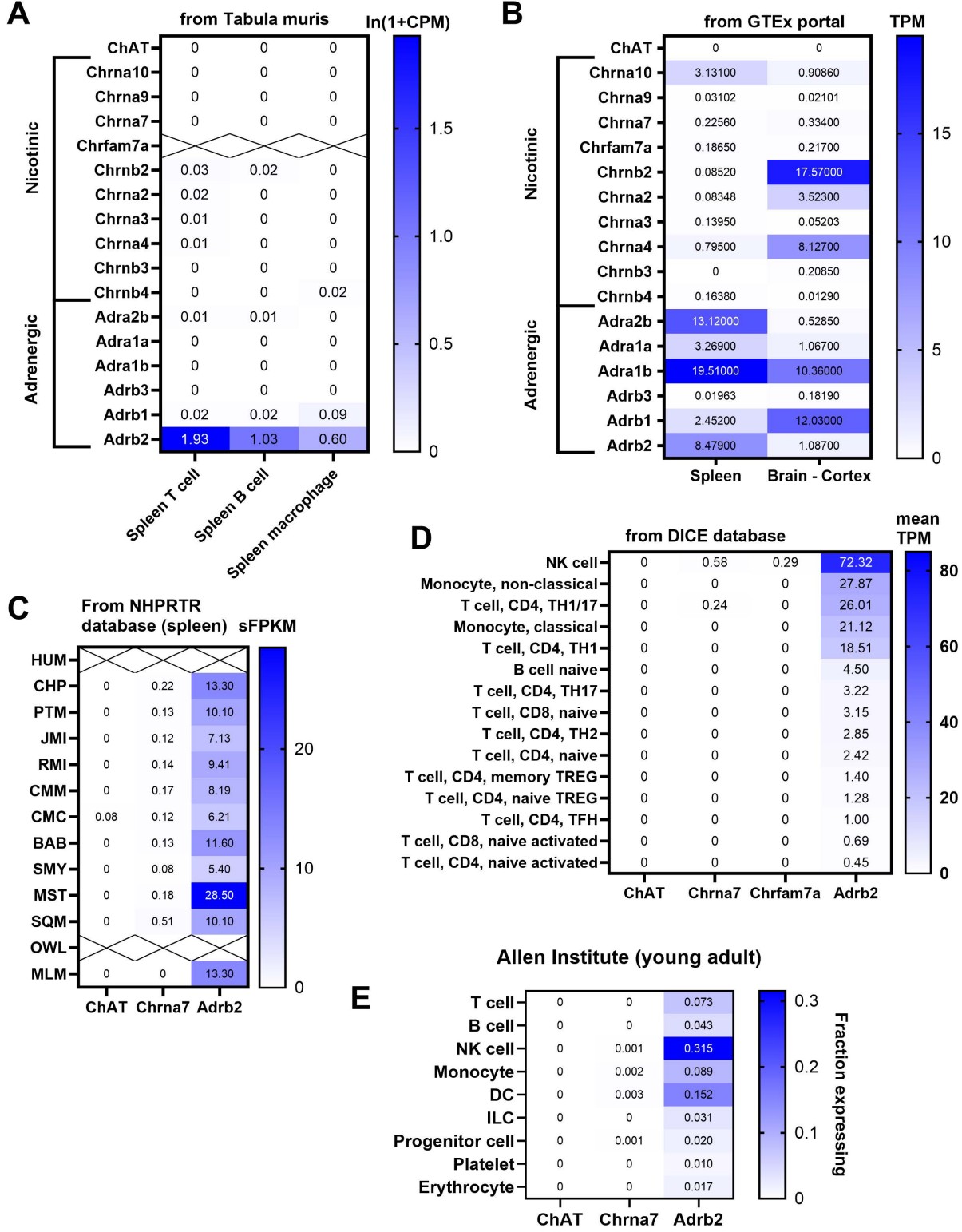

**Fig 8. Heat maps compiling publicly available transcriptomics data from various databases.** Common markers of cholinergic and adrenergic signaling including, but not limited to, *Chrna7*, *Chat*, and *Adrb2*, were assessed across different animal species (mouse, human, non-human primates) and cell types (spleen, brain, cortex, specific immune cell types). **(A)** The Tabula Muris compiles single cell expression profiles from various mouse cell types

including splenic immune cells. Original data can be found at <https://tabula-muris.ds.czbiohub.org/>. **(B)** The GTEx Portal combines RNAseq data obtained from healthy human donors including the spleen. Original data can be found at <https://www.gtexportal.org/home/>. **(C)** The NHPRPR project provides RNAseq data from tissues obtained from humans (HUM) and many non-human primates species. Original data and abbreviations can be found at <https://nhprtr.org/>. **(D)** The DICE database is an RNAseq resource for human immune cells. Original data can be found at <https://dice-database.org/>. **(E)** The Allen Institute offers a collection of single cells data of cell types including immune cells from young and healthy human donors. Original data can be found at <https://apps.allenimmunology.org/aifi/resources/imm-health-atlas>. Raw values are included in each cell. In line with our biochemical and histological findings, splenocytes and immune cells were consistently enriched in adrenergic markers and impoverished in cholinergic markers.

However, it may not come from a local source. In support of this view, our current data and publicly available databases do not provide convincing evidence of any *Chat* expression in the spleen (Fig 8A–8D). Of course, it cannot be ruled out that splenic cells may start producing *Chat* under specific immune challenges and, furthermore, that different animal species or mouse strains express varying levels of Chat.

In contrast, *Adrb2* is highly expressed in the mouse spleen, consistent with numerous previous reports [93] (see also Fig 8). Many *in vivo* studies have demonstrated that Adrb2 signaling plays a significant role in inhibiting inflammation across various paradigms, including, but not limited to, septic shock [94–97]. Here, LPS administration down regulated *Adrb2* expression in the spleen, further highlighting the role of adrenergic signaling in immunomodulation in response to LPS. Although it remains unclear whether LPS-induced downregulation of *Adrb2* has been previously demonstrated *in vivo*, at least one prior *in vitro* study reported that LPS treatment of cultured macrophages reduces *Adrb2* expression [98]. Furthermore, this LPS-induced downregulation of *Adrb2* occurred in a TRIF-dependent manner [98]. The reduced expression of *Adrb2* in α7nAChR knockout mice is an intriguing finding. The underlying cause of this downregulation is unknown, but it is possible that the absence of α7nAChR leads to elevated background inflammation and/or altered immune cells composition, which in turn may suppress *Adrb2* expression.

If splenocytes and immune cells neither express α7nAChRs nor produce Ach, then how to explain the so-called anti-inflammatory cholinergic pathway? Even if our observations do not provide functional insights into how the vagus nerve modulates immunity, they call for a reappraisal of the cellular and anatomical determinants of the anti-inflammatory cholinergic pathway (Fig 9). Considering the enrichment of *Chna7* in many autonomic neurons and its scarcity in the spleen, our data suggest that the anti-inflammatory cholinergic pathway may mainly rely on extrasplenic neuronal α7nAChRs.

Prior studies have shown that both vagal and spinal afferents are highly sensitive to ACh and nicotine [66,81,99–102]. Therefore, α7nAChRs on sensory terminal endings is likely to contribute to the firing of spinal afferents after the electrical stimulation of vagal motor neurons. In parallel, stimulated enteric neurons will likely trigger various mechanical and endocrine responses which may further enhance peripheral afferents activity (Fig 9). For instance, Komegae and colleagues showed that the mechanical manipulation of abdominal viscera stimulates vagal afferent neurons and causes an anti-inflammatory effect [103]. Hence, the electrical stimulation of vagal efferents, which has been conventionally described as sparing afferents, is likely to excite a variety of sensory endings including those located on the side contralateral to the electrical stimulation (Fig 9). Put it briefly, our proposed model suggests that α7nAChRs-bearing peripheral afferents constitute an intermediate link between vagal motor stimulation and inhibited inflammation by the sympathetic system. In particular, one of the most powerful anti-inflammatory pathway in the body involves the sympathetic outflow to the spleen [104]. Once peripheral afferents are stimulated, then a multisynaptic neural pathway enhances the sympathetic outflow to the spleen and other abdominal organs, ultimately inhibiting immunity via the β-2 adrenergic receptor [8,15,50,53,104–109] (Fig 9). The link between sympathetic outflow and immunity is otherwise well established and has continued to be an active area of research for several decades [50,91,104,106,110].

In conclusion, numerous uncertainties remain as to the mechanisms of action and efficacy of the anti-inflammatory cholinergic pathway. Based on the current data, we propose that no irrefutable evidence exists supporting direct cholinergic

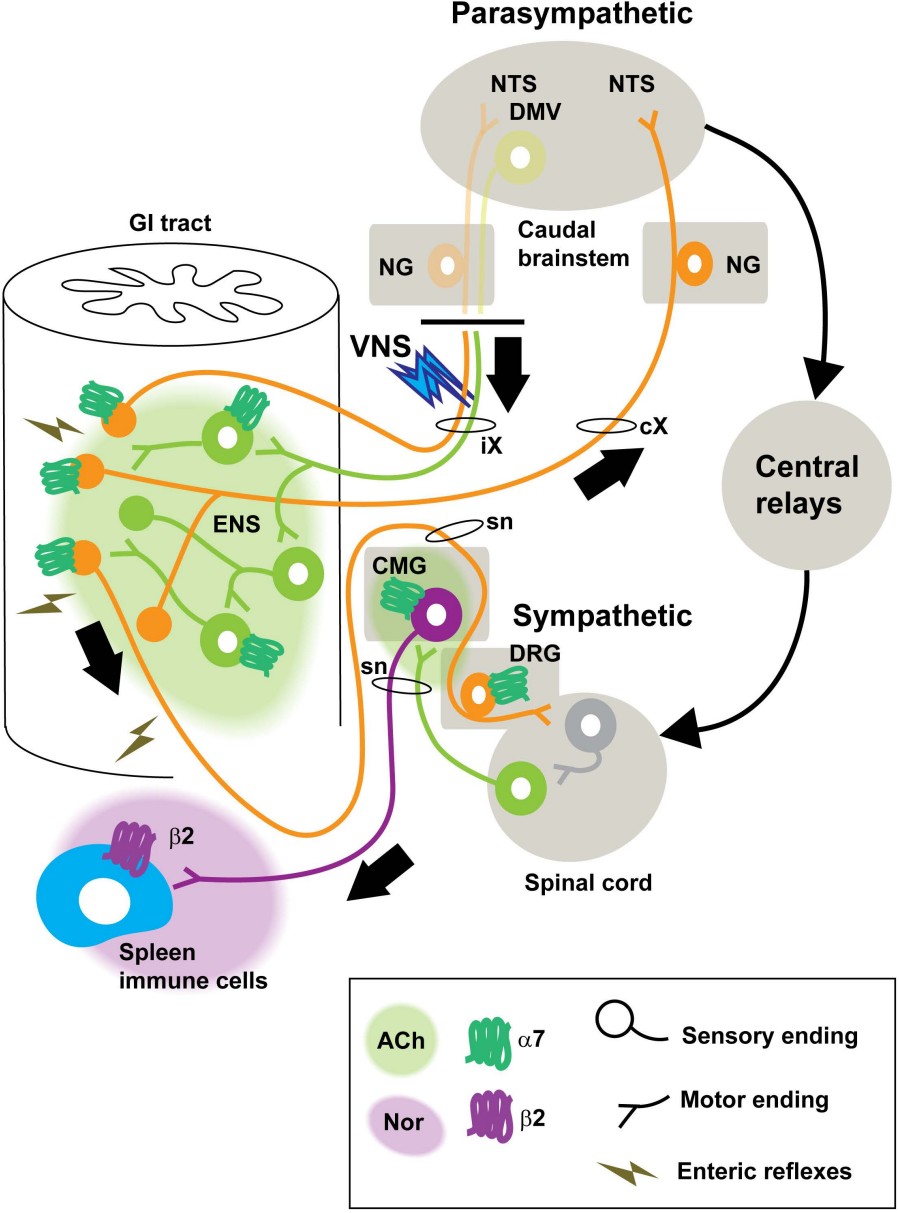

**Fig 9. Proposed alternative model of the anti-inflammatory cholinergic pathway.** We speculate that the unilateral electrical stimulation of the peripheral end of the vagus nerve (VNS) triggers a cascade of nerve reflexes on both sides of the body. Upon the release of acetylcholine from enteric neurons, vagal efferents, and epithelial cholinergic cells, stimulation of α7nAChR-bearing sensory neurons (vagal and spinal) endings occurs. In turn, the stimulation of gut-to-brain pathways mobilizes the sympathetic outflow to the gut and spleen. Ultimately, adrenergic signaling in peripheral immune cells may contribute to suppressing immunity. The expression of β2 receptors in immune cells is otherwise well-established. For the sake of simplicity, the adrenergic supply to the gut is not indicated. In this model, and in agreement with others [51,105], cholinergic signaling in peripheral immune cells themselves plays little or no role in mediating the anti-inflammatory effects of vagus nerve stimulation. Here, we found little evidence of α7nAChR expression in the mouse spleen. Further studies are warranted to validate our speculative model including, most notably, studies with mice that lack α7nAChR only in neurons. If our hypothesis is correct, we predict such animals to be unresponsive to the anti-inflammatory actions of vagus nerve stimulation and α7nAChRs agonists. Black arrows indicate the direction of electrophysiological impulses. Abbreviations: ACh, acetylcholine; CMG, celiac-mesenteric ganglion; cX, contralateral vagus nerve; DMV, dorsal motor of the vagus; DRG, dorsal root ganglion; ENS, enteric nervous system; NG, nodose ganglion; NTS, nucleus of solitary tract; Nor, norepinephrine; iX, ipsilateral vagus nerve; sn, spinal nerves; VNS, vagus nerve stimulation; GI, gastrointestinal; α7, alpha 7 nicotinic acetylcholine receptors.

signaling in splenic cells. Moving forward, research on the neural control of immunity should focus on how cholinergic signaling in extrasplenic sites, including autonomic neurons, rather than immune cells, inhibit inflammation.

## Supporting information

**S1 Fig. Table of qPCR data.**
(DOCX)

## Acknowledgments

We are grateful to Jenny Lee for her assistance with RNAscope studies in the early phase of this project. This basic research study was not pre-registered.

## Author contributions

**Conceptualization:** Steven Wyler, Laurent Gautron.

**Data curation:** Bandy Chen, Laurent Gautron.

**Formal analysis:** Warda Merchant, Bandy Chen, Laurent Gautron.

**Funding acquisition:** Laurent Gautron.

**Investigation:** Warda Merchant, Bandy Chen, Steven Wyler, Laurent Gautron.

**Methodology:** Warda Merchant, Laurent Gautron.

**Project administration:** Laurent Gautron.

**Resources:** Steven Wyler.

**Supervision:** Laurent Gautron.

**Visualization:** Laurent Gautron.

**Writing – original draft:** Laurent Gautron.

**Writing – review & editing:** Laurent Gautron.

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
