## [Decision Letter · Decision Letter 0]

6 May 2025

PONE-D-25-18446
The Molecular Components of The Anti-Inflammatory Cholinergic Pathway Are Extrasplenic
PLOS ONE

Dear Dr. Gautron,

Thank you for submitting your manuscript to PLOS ONE. After careful consideration, we feel that it has merit but does not fully meet PLOS ONE’s publication criteria as it currently stands. Therefore, we invite you to submit a revised version of the manuscript that addresses the points raised during the review process.

We look forward to receiving your revised manuscript.

Kind regards,

Yoshihiko Kakinuma, M.D., Ph.D

Academic Editor

PLOS ONE

Journal Requirements:

“LG: NIH P01DK119130 (CNS mechanisms linking exercise training with energy balance and metabolism, Core C). “

“The NIH grant P01DK119130 (CNS mechanisms linking exercise training with energy balance and metabolism, Core C). The Zeiss LSM880 with Airyscan was purchased with a Shared Instrumentation grant from NIH award 1S10OD021684-01 to Katherine Luby Phelps (UTSW). We are grateful to Jenny Lee for her assistance with RNAscope studies in the early phase of this project. This basic research study was not pre-registered.”

“LG: NIH P01DK119130 (CNS mechanisms linking exercise training with energy balance and metabolism, Core C). “

**Additional Editor Comments:**

The manuscript has been reviewed by two referees, both of which request to revise it in terms of each comments and critisism. They specifically address the request to re-evaluate the accuracy regarding the conclusion. Since the spleen does neither express Chat mRNA nor alpha7 nAChR at all in residential immune cells, and therefore, it should be logical that ACh does not come from the spleen, rather from the vagus nerve. However, if so, vagus nerve stimulation may elevate local AChs in several tissues other than the spleen; moreover, even the splenectomized mice subjected to LPS load could be rescued by the stimulation as well as non-splenectomized mice. Based on referees' and editor's comments they need to conduct another experiment in order to consolidate their speculation and conclusion.

Reviewers' comments:

Reviewer's Responses to Questions

**Comments to the Author**

1. Is the manuscript technically sound, and do the data support the conclusions?

Reviewer #1: Yes

Reviewer #2: Partly

2. Has the statistical analysis been performed appropriately and rigorously? 

Reviewer #1: Yes

Reviewer #2: Yes

3. Have the authors made all data underlying the findings in their manuscript fully available?

Reviewer #1: Yes

Reviewer #2: Yes

4. Is the manuscript presented in an intelligible fashion and written in standard English?

Reviewer #1: Yes

Reviewer #2: Yes

5. Review Comments to the Author

Reviewer #1: The manuscript is well written. Data are clear. This reviewer has a few comments on this manuscript.

The authors proposed the alternative model of the anti-inflammatory cholinergic pathway (shown in Figure 9) based on their mRNA expression data alone. Since no functional investigations were performed, the discussion section appears a bit speculative.

The conclusions are based solely on mRNA expression data. Since detectable mRNA levels do not always correlate with protein abundance, it would be interesting to assess protein expression to strengthen the authors’ proposal.

Reviewer #2: The author concerned anti-inflammatory cholinergic pathway interacted between cholinergic vagal nerves and splenic immune cells so they proposed a new model of the cholinergic anti-

inflammatory pathway that highlights the roles of extrasplenic cholinergic signaling. This is a very interesting paper but there are major issues as shown below.

Major points

The manuscript needs to be reorganized and rewritten to ensure logical and appropriate placement of all content. Some information is presented in inappropriate sections while missing from where it would be most relevant.

The descriptions of statistical analysis are very poor. The information of statistical analysis is need to provide evidence of statistical significance (determination of p-value, F-value, main effect, interaction, etc.) in the methods and the result sections not in the figure legend.

The manuscript lacks logical coherence in the conclusion and the experimental results are not discussed in sufficient depth to support the conclusions. For example, what mechanism about LPS-induced decreasing of the Adrb2 mRNA expression in the brainstem and spleen? The interpretation of the results is needed in the discussion section. There is insufficient interpretation or inference based on the presented data in the discussion section. It might be helpful to rewrite the section by linking each experimental result with previous reports to build a logical hypothesis, and then summarize the overall findings using Figures 8 and 9 in the discussion section.

Only findings from present study should be described in the conclusions and references may be avoided in the conclusion section.

The description in the figure 8 lacks of specific explanation each figure (A-E). Readers need more details of the figure not URL.

Some of the description in the figure 9 is not the explanation of the figure. The author should reorganize the figure legend.

LPS-induced decreasing gene expression of Adrb2 in the brainstem and spleen were well established by qPCR analysis in wild type mice. Several experiments are needed to explain that Chrna7 and Chat mRNA expression is not influenced by LPS injection (several dose of LPS and long-time course).

Fluorescent RNAScope assays revealed that the expression of Chrna7 using is restricted to neurons. The expression of Chrna7 in the afferent neurons is very interesting, so the description of Chrna7 mRNA expression in the vagal and spinal afferent neuron and the involvement of these neurons in the anti-inflammatory pathway is need in the discussion section.

In the result section about the figure 5, the description “the expression of Chrna7 mRNA in the spleen was not significantly different between genotypes, we concluded low levels of Chrna7 in the spleen are attribute to a slight sample contamination during dissection an/or tissue processing.” is inappropriate. What means sample contamination? The phrases impress reduced reliability of the experimental data. I recommend verifying reproducibility with large sample size of mice.

The expression of Chrna7 is not detected by using RNAScope assay, so the reason about this discrepancy is needed in the discussion.

Minor points

How about ChAT-positive cell body in the nodose ganglion?

In the figure 6, the authors did not mention the ChAT-positive neuronal cell body may be exist in the nodose ganglion. To ensure the validity of the study, it should be mentioned in the result section.

According to submission guideline, in the text, the reference number cite in square brackets (e.g., “We used the techniques developed by our colleagues [19] to analyze the data”). You must change the description of the reference number.

The words “vagal afferents” is not suitable, so change to “vagal afferent neurons” in the line 199.

In the legend of Figure 6, what is mean of the description “saline and LPS”?

Some description in the figure legend contains the appropriate content for the methods, so move and rewrite in the appropriate section.

The word “mRNA” is mentioned in the description about data from qPCR and RNA scope.

Change “Chrn7” to “Chrna7” in the page18, line379, 381, 383, 386, 387, 386.

Change “microscopy” to “microscopic analysis” in the page18 line392.

6. PLOS authors have the option to publish the peer review history of their article (what does this mean?). If published, this will include your full peer review and any attached files.

Reviewer #1: No

Reviewer #2: No

---

## [Author Response · Author response to Decision Letter 1]

10 Jul 2025

Reviewer #1: The manuscript is well written. Data are clear. This reviewer has a few comments on this manuscript.

The authors proposed the alternative model of the anti-inflammatory cholinergic pathway (shown in Figure 9) based on their mRNA expression data alone. Since no functional investigations were performed, the discussion section appears a bit speculative.

Answer: Thank you for the helpful and constructive feedback. We agree that our discussion was too speculative and wordy. Therefore, we revised it to remove entirely the speculative parts not directly linked to our own data.

The conclusions are based solely on mRNA expression data. Since detectable mRNA levels do not always correlate with protein abundance, it would be interesting to assess protein expression to strengthen the authors’ proposal.

Answer: We appreciate the suggestion to assess the α7 nicotinic acetylcholine receptor (α7 nAChR) protein. While this would indeed be of interest, to our knowledge, there are currently no specific and well-characterized antibodies available for this receptor. Unfortunately, the reliability of most commercially available antibodies targeting G protein-coupled receptors (GPCRs), including α7, remains a significant concern. This issue has been extensively documented by us and others in the field (see PMID: 37264690; 31080409). ChAT protein has been examined in the past by many investigators in the past and we do not wish to repeat this information. While there are reliable antibodies against ChAT, immunodetection of ChAT remains challenging, which is also an issue we discussed in detail in the past (PMID: 23749724; 34729780). In other words, we feel strongly that examining proteins is simply not feasible now.

Reviewer #2: The author concerned anti-inflammatory cholinergic pathway interacted between cholinergic vagal nerves and splenic immune cells so they proposed a new model of the cholinergic anti-

inflammatory pathway that highlights the roles of extrasplenic cholinergic signaling. This is a very interesting paper but there are major issues as shown below.

Answer: Thank you for the helpful and constructive feedback. We addressed your concerns to the further extent possible as explained below. In particular, we extensively revised our discussion which both reviewers found too speculative.

Major points

The manuscript needs to be reorganized and rewritten to ensure logical and appropriate placement of all content. Some information is presented in inappropriate sections while missing from where it would be most relevant.

Answer: We apologize for the lack of clarity. Please find a revised manuscript which hopefully addresses your concerns (see below for details).

The descriptions of statistical analysis are very poor. The information of statistical analysis is need to provide evidence of statistical significance (determination of p-value, F-value, main effect, interaction, etc.) in the methods and the result sections not in the figure legend.

Answer: We agree to reorganize our results according to reviewer #2’s comments. Results with statistics were moved from legends to the main manuscript. We are grateful to Reviewer #2 for pointing out the issues.

The manuscript lacks logical coherence in the conclusion and the experimental results are not discussed in sufficient depth to support the conclusions. For example, what mechanism about LPS-induced decreasing of the Adrb2 mRNA expression in the brainstem and spleen? The interpretation of the results is needed in the discussion section. There is insufficient interpretation or inference based on the presented data in the discussion section.

Answer: We agree and our revised manuscript discusses potential mechanisms through which LPS regulated Adrb2 based on the available literature (see PMID: 19167076; see also our revised discussion).

 It might be helpful to rewrite the section by linking each experimental result with previous reports to build a logical hypothesis, and then summarize the overall findings using Figures 8 and 9 in the discussion section.

Answer: Our manuscript was revised to be clearer, less speculative, and more logical.

Only findings from present study should be described in the conclusions and references may be avoided in the conclusion section.

Answer: We extensively revised our discussion to focus on our own findings as suggested by Reviewer #2.

The description in the figure 8 lacks of specific explanation each figure (A-E). Readers need more details of the figure not URL.

Answer: We agree and we revised figure 8 accordingly.

Some of the description in the figure 9 is not the explanation of the figure. The author should reorganize the figure legend.

Answer: We agree and we revised figure 9 accordingly.

LPS-induced decreasing gene expression of Adrb2 in the brainstem and spleen were well established by qPCR analysis in wild type mice. Several experiments are needed to explain that Chrna7 and Chat mRNA expression is not influenced by LPS injection (several dose of LPS and long-time course).

Answer: Given that the anti-inflammatory effects of vagus nerve stimulation (VNS) manifest rapidly—typically within minutes to a few hours—we did not deem it necessary to assess the time course of Chrna7 expression over an extended period. The time point selected in our study (1.5 hours post-LPS) aligns with that used in the seminal work describing the cholinergic anti-inflammatory reflex (2 hours post-LPS; PMID: 12508119). Furthermore, it is important to note that multiple studies have demonstrated that VNS can suppress inflammation when applied prior to LPS administration (see discussion in PMID: 30396599). This strongly suggests that Chrna7 and Chat must already be expressed prior to immune challenge. Cells expressing α7 nicotinic receptors and capable of releasing acetylcholine are therefore expected to be present at baseline, before LPS administration.

Importantly, the anti-inflammatory cholinergic pathway was originally described using a much lower dose of LPS (0.1 mg/kg, i.p.) by Wang et al. (PMID: 12508119). Therefore, it is unlikely that the moderate LPS dose (1 mg/kg, i.p.) used in our experiments was insufficient to induce Chat expression in the spleen. However, we agree that it is scientifically relevant to explore whether a very high dose of LPS could influence Chat expression. To address this point, we conducted additional qPCR control experiments using a very high LPS dose (see new supplementary data). Notably, Chat remained undetectable regardless of the dosage.

Fluorescent RNAScope assays revealed that the expression of Chrna7 using is restricted to neurons. The expression of Chrna7 in the afferent neurons is very interesting, so the description of Chrna7 mRNA expression in the vagal and spinal afferent neuron and the involvement of these neurons in the anti-inflammatory pathway is need in the discussion section.

Answer: We agree with the reviewer. We hope that our revised version discusses in enough detail the issue of neuronal Chrna7.

In the result section about the figure 5, the description “the expression of Chrna7 mRNA in the spleen was not significantly different between genotypes, we concluded low levels of Chrna7 in the spleen are attribute to a slight sample contamination during dissection an/or tissue processing.” is inappropriate. What means sample contamination? The phrases impress reduced reliability of the experimental data. I recommend verifying reproducibility with large sample size of mice. The expression of Chrna7 is not detected by using RNAScope assay, so the reason about this discrepancy is needed in the discussion.

Answer: We apologize for our poor choice of words (e.g. contamination). We rewrote our text to avoid further confusion. We did not mean to say that our samples were technically “contaminated”, but that residual Chrna7 detection by qPCR may be due to a very small number of cells that are not undetectable by histology.

Minor points

How about ChAT-positive cell body in the nodose ganglion? In the figure 6, the authors did not mention the ChAT-positive neuronal cell body may be exist in the nodose ganglion. To ensure the validity of the study, it should be mentioned in the result section.

Answer: This is a good question. As noted in our Chat-GFP data, we did not see any Chat-positive cell bodies in the nodose ganglion, but only axons. This is now better explained in our revision.

According to submission guideline, in the text, the reference number cite in square brackets (e.g., “We used the techniques developed by our colleagues [19] to analyze the data”). You must change the description of the reference number. Answer:

We modified the formatting of our references.

The words “vagal afferents” is not suitable, so change to “vagal afferent neurons” in the line 199.

Answer: We agree and modified our text.

In the legend of Figure 6, what is mean of the description “saline and LPS”?

Answer: We apologize for the mistake and modified the manuscript.

Some description in the figure legend contains the appropriate content for the methods, so move and rewrite in the appropriate section.

Answer: We modified our text.

The word “mRNA” is mentioned in the description about data from qPCR and RNA scope.

Answer: We agree and modified our text.

Change “Chrn7” to “Chrna7” in the page18, line379, 381, 383, 386, 387, 386.

Answer: Thank you for catching these errors. We corrected our text.

Change “microscopy” to “microscopic analysis” in the page18 line392.

Answer: We agree and modified our text.

---

## [Decision Letter · Decision Letter 1]

29 Jul 2025

PONE-D-25-18446R1
The Molecular Components of The Anti-Inflammatory Cholinergic Pathway Are Extrasplenic
PLOS ONE

Dear Dr. Gautron,

Thank you for submitting your manuscript to PLOS ONE. After careful consideration, we feel that it has merit but does not fully meet PLOS ONE’s publication criteria as it currently stands. Therefore, we invite you to submit a minorly revised version of the manuscript that addresses the points raised during the review process.
 
The authors have almost revised original version according to the referees suggestion. however, 
they have left several points unrevised despite the Referee 2 concerns., especially, 1) p values in figure legends depicted by asterisk not yet added, 2) references in the conclusion sentence, and 3) several typos in methods (Line 86, 131, and 246 in the revised version). Therefore, they need to revise them.

We look forward to receiving your revised manuscript.

Kind regards,

Yoshihiko Kakinuma, M.D., Ph.D

Academic Editor

PLOS ONE

Journal Requirements:

Additional Editor Comments:

The authors extensively revised the original manuscript based on the referees comments, and the speculative parts are fully revised. However, several parts which the referee pointed out are left not revised. Moreover, several typos have been still observed. For example, 1) Line 86, "since all our procedures were terminal. are listed". Does this mean that "since our procedures were terminal and are listed"?; 2) Line 131, "Prevalidated taqman assays were Mm01317884_ml (Chrna7)" Does this mean that "Prevalidated Taqman assays were performed using primers of Mm01317884_ml (Chrna7)"?; 3) Line 246, "(72)" should be altered to [72]. In addition, the authors need to re-revise the manuscript under the consideration of the referee 2 suggestions including statistical explanation in figure legends (p values) and deletion of references in a conclusion sentences.

Reviewers' comments:

Reviewer's Responses to Questions

**Comments to the Author**

1. If the authors have adequately addressed your comments raised in a previous round of review and you feel that this manuscript is now acceptable for publication, you may indicate that here to bypass the “Comments to the Author” section, enter your conflict of interest statement in the “Confidential to Editor” section, and submit your "Accept" recommendation.

Reviewer #1: All comments have been addressed

Reviewer #2: (No Response)

2. Is the manuscript technically sound, and do the data support the conclusions?

Reviewer #1: Yes

Reviewer #2: Partly

3. Has the statistical analysis been performed appropriately and rigorously? 

Reviewer #1: Yes

Reviewer #2: Yes

4. Have the authors made all data underlying the findings in their manuscript fully available?

Reviewer #1: Yes

Reviewer #2: Yes

5. Is the manuscript presented in an intelligible fashion and written in standard English?

Reviewer #1: Yes

Reviewer #2: Yes

6. Review Comments to the Author

Reviewer #1: Although this reviewer still has an interest in the protein expression, this reviewer understood the author's concerns regarding antibodies.

This reviewer considers the current version of the manuscript to be acceptable for publication.

Reviewer #2: Comments to the Author

The authors have addressed majority of the concerns and the manuscript is substantially improved. I especially appreciate the manuscript was revised according to our concerns.

There are several concerns remaining:

Major points

As I mentioned in my previous comment, it seems that it has not been fully understood. The following two points are presented below.

1. The author must read “PLOS ONE Statistical Reporting Guidelines“ carefully and revise the statistical analysis in the material and method, results and figure legend. Information about which statistical methods were used for which data, and which parameters were used to assess the effects with ANOVA, should be described in the methods section. Although statistical results are revised and presented in the results section, the definition of asterisks should be included in the figure legend.

2. Descriptions of inferences and interpretations are not appropriate for the results section. Please move these statements found in the results section to the discussion section. The results section should contain only the results and no other content.

Minor points

As I mentioned before, citations are not necessary in the Conclusion section.

7. PLOS authors have the option to publish the peer review history of their article (what does this mean?). If published, this will include your full peer review and any attached files.

Reviewer #1: No

Reviewer #2: No

---

## [Author Response · Author response to Decision Letter 2]

13 Aug 2025

Response letter

“The authors have almost revised original version according to the referees suggestion. however,

they have left several points unrevised despite the Referee 2 concerns., especially, 1) p values in figure legends depicted by asterisk not yet added, 2) references in the conclusion sentence, and 3) several typos in methods (Line 86, 131, and 246 in the revised version). Therefore, they need to revise them.”

“Additional Editor Comments:

The authors extensively revised the original manuscript based on the referees comments, and the speculative parts are fully revised. However, several parts which the referee pointed out are left not revised. Moreover, several typos have been still observed. For example, 1) Line 86, "since all our procedures were terminal. are listed". Does this mean that "since our procedures were terminal and are listed"?; 2) Line 131, "Prevalidated taqman assays were Mm01317884_ml (Chrna7)" Does this mean that "Prevalidated Taqman assays were performed using primers of Mm01317884_ml (Chrna7)"?; 3) Line 246, "(72)" should be altered to [72]. In addition, the authors need to re-revise the manuscript under the consideration of the referee 2 suggestions including statistical explanation in figure legends (p values) and deletion of references in a conclusion sentences.”

Dear editors,

We addressed previously raised concerns as explained below. Changes are marked-up in our revised manuscript. All typos were corrected. In addition, P values corresponding to the asterisks in each graph were added to the legends.

“Review Comments to the Author

Reviewer #1: Although this reviewer still has an interest in the protein expression, this reviewer understood the author's concerns regarding antibodies.

This reviewer considers the current version of the manuscript to be acceptable for publication.

We thank Reviewer 1 for his/her helpful comments throughout the revision process.

Reviewer #2: Comments to the Author

The authors have addressed majority of the concerns and the manuscript is substantially improved. I especially appreciate the manuscript was revised according to our concerns.

We thank Reviewer 2 for his/her helpful comments throughout the revision process.

There are several concerns remaining:

Major points

As I mentioned in my previous comment, it seems that it has not been fully understood. The following two points are presented below.

1. The author must read “PLOS ONE Statistical Reporting Guidelines“ carefully and revise the statistical analysis in the material and method, results and figure legend. Information about which statistical methods were used for which data, and which parameters were used to assess the effects with ANOVA, should be described in the methods section. Although statistical results are revised and presented in the results section, the definition of asterisks should be included in the figure legend.

As requested by reviewer 2, we added a section describing our statistical reporting.

2. Descriptions of inferences and interpretations are not appropriate for the results section. Please move these statements found in the results section to the discussion section. The results section should contain only the results and no other content.

We deleted interpretations and comments from the results sections whenever appropriate.

Minor points

As I mentioned before, citations are not necessary in the Conclusion section.”

We apologize for the misunderstanding. We have removed the citations included in the conclusion sentence.

---

## [Decision Letter · Decision Letter 2]

20 Aug 2025

The Molecular Components of The Anti-Inflammatory Cholinergic Pathway Are Extrasplenic

PONE-D-25-18446R2

Dear Dr. Laurent Gautron,

We’re pleased to inform you that your manuscript has been judged scientifically suitable for publication and will be formally accepted for publication once it meets all outstanding technical requirements.

Kind regards,

Yoshihiko Kakinuma, M.D., Ph.D

Academic Editor

PLOS ONE

Additional Editor Comments (optional):

Reviewer 1 has already recommended 'accept', after evaluating the 1st revised manuscript. On the other hand, the Reviewer 2 has recognized that the 2nd revised manuscript is suitable for 'accept'.

Based on the two reviewers' comments, the authors have addressed almost all comments and concerns from reviewers. Therefore, nothing has been left for further evaluation.

Reviewers' comments:

Reviewer's Responses to Questions

**Comments to the Author**

1. If the authors have adequately addressed your comments raised in a previous round of review and you feel that this manuscript is now acceptable for publication, you may indicate that here to bypass the “Comments to the Author” section, enter your conflict of interest statement in the “Confidential to Editor” section, and submit your "Accept" recommendation.

Reviewer #2: All comments have been addressed

2. Is the manuscript technically sound, and do the data support the conclusions?

Reviewer #2: Yes

3. Has the statistical analysis been performed appropriately and rigorously? 

Reviewer #2: Yes

4. Have the authors made all data underlying the findings in their manuscript fully available?

Reviewer #2: Yes

5. Is the manuscript presented in an intelligible fashion and written in standard English?

Reviewer #2: Yes

6. Review Comments to the Author

Reviewer #2: (No Response)

7. PLOS authors have the option to publish the peer review history of their article (what does this mean?). If published, this will include your full peer review and any attached files.

Reviewer #2: No

---

## [Editor Report · Acceptance letter]

PONE-D-25-18446R2

PLOS ONE

Dear Dr. Gautron,

I'm pleased to inform you that your manuscript has been deemed suitable for publication in PLOS ONE. Congratulations! Your manuscript is now being handed over to our production team.

Kind regards,

on behalf of

Professor Yoshihiko Kakinuma

Academic Editor

PLOS ONE